# PETRI: Learning Unified Cell Embeddings from Unpaired Modalities via Early-Fusion Joint Reconstruction

**Ryan Conrad, Ethan Weinberger, Saradha Venkatachalapathy, Owen Chen**
**Darshini Shah, Bay Johnson, Max R Salick, Vaishaali Natarajan & Emily B. Fox**
insitro
South San Francisco, CA 94080, USA
`{rconrad,,ethan.weinberger,saradhavpathy,owen,darshini,bay,`
`max,vaishaali,emily}@insitro.com`

## Abstract

Integrating imaging and transcriptomics screening data holds promise for isolating true biological signals from modality-specific technical artifacts. However, existing multimodal embedding approaches either require pairing or fail to capture both shared and modality-specific information in an end-to-end manner. We present PETRI, an early-fusion transformer that learns a unified cell embedding from unpaired cellular images and gene expression profiles. PETRI groups cells by shared experimental context into multimodal "documents" and performs masked joint reconstruction with cross-modal attention, permitting information sharing while preserving modality-specific capacity. The resulting latent space supports construction of perturbation-level profiles by simple averaging across modalities. Applying sparse autoencoders to the embeddings reveals learned concepts that are biologically meaningful, multimodal, and retain perturbation-specific effects. To support further machine learning research, we release a blinded, matched optical pooled screen (OPS) and Perturb-seq dataset in HepG2 cells.

## 1 Introduction

A major goal of cell biology is to establish causal links between gene activity and cellular states (Rood et al., 2024). High-throughput perturbation technologies now profile complementary facets of these states at scale. Perturb-seq (Dixit et al., 2016) combines CRISPR-based perturbations with single-cell RNA sequencing to read out transcriptome-wide effects, while Optical Pooled Screening (OPS) (Feldman et al., 2019; Sivanandan et al., 2023; Ramezani et al., 2025) uses cost-effective fluorescence microscopy to capture morphological phenotypes. Together, transcriptomics and imaging provide complementary views of how perturbations reshape cellular state.

The growing availability of large Perturb-seq and OPS datasets motivates methods for multimodal representation learning that capture their shared and modality-specific information. Shared structure can help disentangle true biological signal from technical confounders—such as random gene dropout in RNA-seq or intensity fluctuations in microscopy—that are unlikely to be supported across modalities (Radhakrishnan et al., 2023). At the same time, modality-specific signals contain unique phenotypes and mechanistic clues that, when combined, yield a more complete picture of perturbation effects.

However, integrating information across these modalities is challenging. First, obtaining data paired at the cell-level is not possible as state-of-the-art assays are destructive and cannot profile both morphology and gene expression from the same cells. Second, morphology and gene expression only partially overlap in the biology that they capture (Way et al., 2022), so models must remain robust when signals are disjoint.

To address these challenges, we present PETRI, an early-fusion self-supervised transformer that learns single-cell embeddings from unpaired images and transcriptomes. PETRI groups cells by

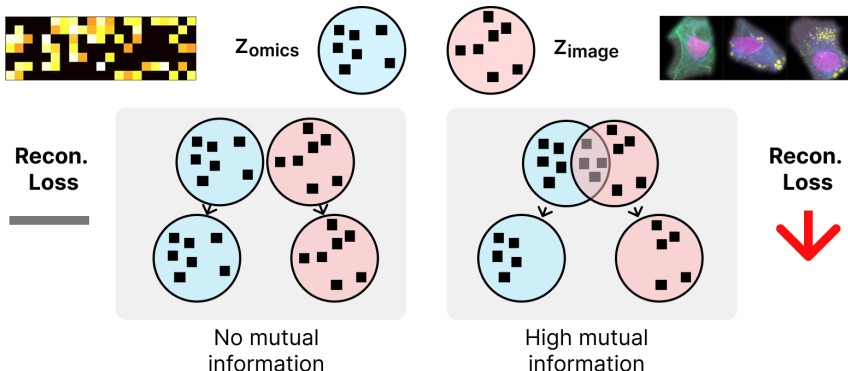

**Figure 1:** Conceptual overview: Joint reconstruction of masked regions promotes multimodal integration. Left: With disjoint modalities, loss matches unimodal decoders. Right: With high mutual information, cross-modal predictions reduce reconstruction loss.

perturbation into multimodal "documents" and employs self-attention to propagate useful features across cells and between modalities.

We show that PETRI learns effective multimodal cellular representations. Perturbation-level embeddings, formed by averaging PETRI cell embeddings across modalities, recover known gene-gene relationships. Ablation analysis and an investigation of cross-modality attention patterns shows that PETRI finds subtle multimodal features that decrease reconstruction loss. Finally, we use sparse autoencoders to isolate these multimodal concepts and show that they are robust to batch effects and correspond to perturbation-specific morphological and molecular concepts.

To catalyze further research, we release a blinded, matched OPS and Perturb-seq dataset in HepG2 cells spanning 569 CRISPR knockouts in four chemical backgrounds.

## 2 RELATED WORK

**Representation learning for cellular images**   Image-based profiling aims to convert the rich information in microscopy images into quantitative feature vectors for downstream analysis in drug discovery and functional genomics (Chandrasekaran et al., 2021). Early deep learning approaches used weak supervision, training models to predict the experimental treatment (e.g., drug or genetic perturbation) applied to a cell or group of cells (Caicedo et al., 2018). However, this approach is limited by its core assumption that all perturbations produce a morphological change, which is often not the case. More recent work has shifted to self-supervised learning. These methods, including vision transformer-based self-distillation and masked autoencoders, have demonstrated state-of-the-art performance at inferring known biological relationships from images without relying on experimental labels for training (Doron et al., 2023; Kraus et al., 2024; Pham et al., 2025). Perturbation-aggregated embeddings are also used for predicting a compound's mechanism of action, identifying disease-specific phenotypes, and functional gene annotation (Sivanandan et al., 2023).

**Representation learning for single cell transcriptomics**   Generative models have become a cornerstone of single-cell transcriptomics analysis, with pioneering methods like scVI using variational autoencoders to learn a probabilistic latent space that corrects for technical noise and batch effects (Lopez et al., 2018). This paradigm has evolved with the advent of foundation models for biology, which leverage the transformer architecture and pre-training on massive-scale datasets of tens of millions of cells (Cui et al., 2024; Theodoris et al., 2023; Gong et al., 2023; Pearce et al., 2025; Kalfon et al., 2025). Common tasks for these models include automated cell type annotation, integration of datasets from different experiments or technologies, and the prediction of cellular responses to genetic or chemical perturbations.

**Multimodal single cell embeddings**   Integrating the many modalities that can be measured from a single cell is a key challenge in modern biology, as highlighted by community-wide efforts such as the NeurIPS 2021 Multimodal Single-Cell Data Integration Challenge (Lance et al., 2022). For

datasets with modalities paired at the cell level, methods focus on learning a joint representation. For instance, deep generative models like MultiVI learn a probabilistic embedding of paired multi-omic data (Ashuach et al., 2023), while contrastive frameworks like scCLIP align paired chromatin accessibility and gene expression profiles (Xiong et al., 2023). This principle also extends to linking imaging with molecular data, where models like OmiCLIP learn to associate histopathology images with their corresponding spatial transcriptomics profiles (Chen et al., 2025b).

However, collecting paired image and transcriptomics data is often expensive or technically infeasible, making methods that can integrate unpaired data crucial. To address this, some approaches aim to align individual cells across modalities; for example, propensity score alignment leverages shared perturbation labels to estimate a matching between cells in different datasets (Xi & Hartford, 2024). Yang et al. (2021) introduce cross-modal autoencoders which use a two-stage learning process, first fitting a variational autoencoder (VAE) to images and then training a VAE for gene expression with a regularization loss that forces the latent spaces to overlap. Other methods operate at the level of cell populations using weaker supervisory signals. CellCLIP learns a shared embedding space between textual descriptions of perturbations and the sets of cell images resulting from them (Lu et al., 2025). Similarly, MultiMIL uses sample-level labels, such as patient disease status, in a multiple-instance learning (MIL) framework to identify the specific cells in different modalities that are characteristic of that label (Litinetskaya et al., 2024).

**Vision-language models for multimodal representation learning**   Recent advances in VLMs have established powerful architectures for multimodal learning. Some models utilize resampling and cross-attention mechanisms to fuse information from interleaved image and text data (Alayrac et al., 2022), while others, like LLaVA, project image features into the word embedding space and process a unified sequence with a standard self-attention mechanism (Liu et al., 2023). Although most often used for visual question answering, the VLM framework can be adapted for representation learning. For example, MoCa is trained to denoise and reconstruct both image and text inputs simultaneously, enabling it to learn effective bidirectional multimodal embeddings from large, unlabeled datasets (Chen et al., 2025a).

**Matched [1] imaging and gene expression datasets**   The LINCS dataset (Way et al., 2022) includes a library of 1,327 chemical perturbations with Cell Painting and L1000 readouts. L1000 measures 978 mRNA transcripts from bulk samples, though the authors of the dataset report that it suffers from poor reproducibility of perturbation effects. Perturb-FISH (Binan et al., 2025) has matched Perturb-seq and OPS with MERFISH (Chen et al., 2015) for 35 genetic perturbations. MERFISH provides single cell pairing of morphology and mRNA counts but is limited to a few hundred genes. Relatedly, Perturb-Multi (Saunders et al., 2025) is a unique spatial transcriptomics dataset that genetically perturbs mouse liver cells *in vivo* and records MERFISH and protein staining.

## 3   METHOD

For multimodal representation learning, CLIP (Radford et al., 2021) may seem like a natural starting point. But, contrastive methods like CLIP are ill-suited to our scenario since their effectiveness hinges on strong positive pairs distinguishable from a large number of negatives. Our dataset has limited unique treatments (about 2,200) and the modalities have no explicitly overlapping features.

This motivates a shift in perspective. Instead of treating each modality instance as a distinct item in a pair, we draw inspiration from VLMs that operate on mixed-modality documents. A document, like a webpage for example, contains data that is merely aligned by a common topic – the context increases the odds of finding cross-modality associations. PETRI conceives of perturbations as topics and builds documents from sets of cells. Our central hypothesis is that cellular phenotypes visible in both modalities, and enriched under certain perturbations, will supply mutual information that improves the reconstruction of corrupted data (**Fig. 1**). If the modalities have no mutual information or even contradictory information, we expect cross-modality attention to decrease and for reconstruction loss and representation quality to default to the level of unimodal models.

The document approach is appealing, but introduces a significant technical hurdle: an exploding sequence length. A single cell can be represented by hundreds of image patch tokens or thousands

---

[1]Here matched refers to both datasets having the same set of perturbations.

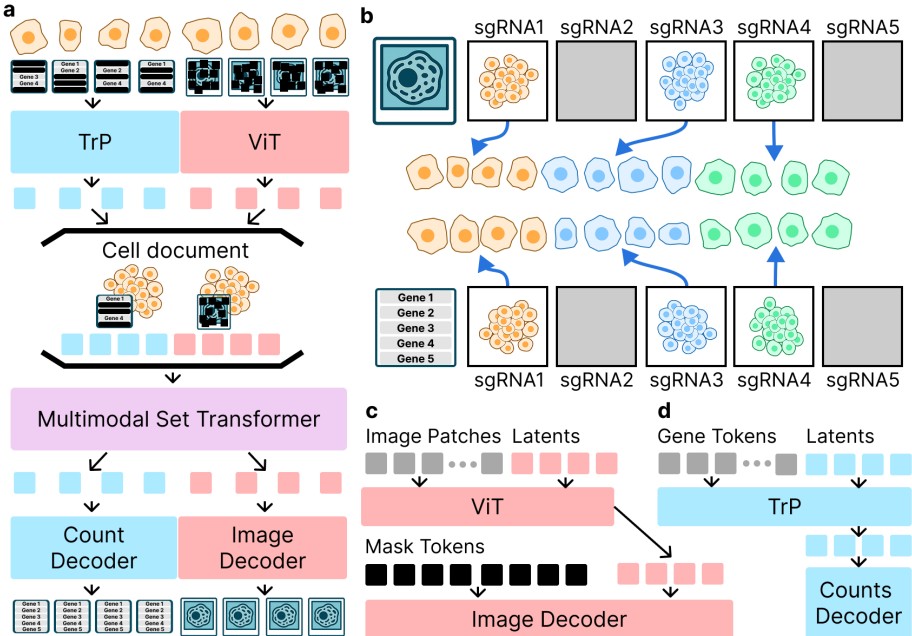

**Figure 2:** The PETRI architecture. **a**, Overview of separate encoders, cell document creation, multimodal set transformer, and separate decoders. **b**, Data grouping and sampling by modality and perturbation. **c**, ViT-based masked autoencoder for images with token resampling. **d**, Perceiver-based masked autoencoder for transcriptomics with token resampling.

of gene tokens. A set of cells would produce a sequence far too long for standard transformers. Our solution is aggressive token resampling. By distilling the token representation of each cell into a small, fixed number of latent tokens, we can flexibly scale the number of cells per document.

**Fig. 2a** outlines PETRI's architecture, which consists of four steps, each detailed in **Sec. 3.1–3.4**:

1. Create batches where cells are grouped by perturbation to form sets (**Fig. 2b**).
2. Convert raw cell data (images or transcriptomes) into tokens, mask a large portion of them, and use a modality-specific encoder to resample the unmasked tokens into a small, fixed-size sequence of latents (**Fig. 2c-d**).
3. Concatenate the latent tokens from cells in the same sets to form multimodal cell documents and process them with a multimodal set transformer.
4. Split the document back into individual cell latent tokens and use modality-specific decoders to reconstruct the original masked input from the latent representations.

## 3.1 BATCH CONSTRUCTION

For training, we stratify datasets into groups based on the experimental context; here we primarily focus on perturbation-level grouping e.g., a unique genetic treatment like the guide RNA (sgRNA), or combination of perturbation and chemical background. From each group, we sample a set of $S$ cells with replacement for each modality. Multiple sets are collated into a mini-batch and dispatched to the appropriate encoder. This process is depicted in **Fig. 2b**.

## 3.2 TOKENIZATION AND PER-CELL RESAMPLING

To construct the multimodal documents, each cell must first be encoded into a compact latent representation. This is achieved with modality-specific encoders designed with token resampling. During training, we randomly mask 75% of input tokens (image patches or genes) per cell and remove them from the sequence.

For images, we follow the standard Vision Transformer (ViT) approach of embedding non-overlapping patches with position encodings. To achieve resampling, we concatenate a fixed number of learnable latent tokens, $L$, to the sequence of image patch tokens, $N$, where $L \ll N$. This combined sequence is processed by transformer blocks, and only the $L$ output latent tokens are retained as the cell's image representation (**Fig. 2c**).

For transcriptomics, the input sequence length, corresponding to thousands of genes, makes a standard transformer architecture computationally infeasible. We therefore required an architecture purpose-built for efficiently processing and resampling extremely long sequences. We adopt the Perceiver (Jaegle et al., 2021), which interleaves cross-attention layers for resampling with self-attention layers over the latent tokens only (**Fig. 2d**). This design directly serves our need for aggressive token resampling, making it a natural architectural choice. Gene expression is tokenized by combining a learned gene embedding with its measured log count via a two-layer MLP. Similar to the image case, we retain $L$ latent tokens as a cell's transcriptomic representation.

### 3.3 MULTIMODAL SET TRANSFORMER (MST)

The modality-specific encoders output a fixed number of latent tokens per cell, resulting in a tensor of shape $(G \times S, L, D)$, where $G$ is the number of groups (e.g., perturbations and/or conditions) in the batch, $S$ is the set size (i.e., number of cells), $L$ is the number of latent tokens, and $D$ is their dimensionality.

To form the cell documents, we reshape this tensor to $(G, S \times L, D)$ and concatenate the representations from both modalities along the token dimension to form a batch of unified sequences with shape $(G, 2 \times S \times L, D)$. These sequences are then processed through a series of standard transformer blocks, allowing for cross-modal and cross-cell attention. Afterwards, we split the sequence by modality and reconstitute the original $(G \times S, L, D)$ shapes for decoding.

### 3.4 DECODERS AND LOSS

The final step is to reconstruct the original inputs from the processed latent tokens, enforcing that the latents capture comprehensive information about each cell.

The image decoder is adapted from Masked Autoencoders (MAEs) (He et al., 2022). Since our latent tokens are not tied to specific patch locations, we concatenate them with a full sequence of $N$ learnable mask tokens. The decoder is trained to reconstruct the original masked image patches from this combined sequence. The loss is the mean squared error (MSE) between the reconstructed and original pixel values of the masked patches only.

For the transcriptomics decoder, we mean-pool the latent tokens for each cell and pass them through a three-layer MLP that outputs a value for each gene. When raw counts are available, we apply a softmax over the gene dimension and use the negative log-likelihood of a negative binomial distribution as the loss. If reconstructing log-normalized counts instead, we use an MSE loss. Analogous to the image modality, the loss is calculated exclusively on masked-out genes.

### 3.5 EVALUATION

We evaluate two metrics on aggregated embeddings from genetic treatment metadata.

**Guide Consistency.** In CRISPR screens, multiple guide RNAs (sgRNAs) are designed to target the same gene for editing, and thus these guides should induce similar phenotypic effects. To assess whether the models' representations are consistent with this prior knowledge, we compute cosine similarities of mean guide embeddings within each target gene and compare them to an empirical null distribution of similarities with the same cardinality between unrelated sgRNAs. The metric we report is the fraction of target genes in the screen with guides that have a statistically significant ($p < 0.05$) similarity after multiple testing correction.

**StringDB edge classification.** Introduced by Sivanandan et al. (2023), this metric uses physically interacting gene-gene pairs collected from the StringDB database as ground truth labels in a zero-shot classification task. Pairwise cosine similarities are computed for aggregated target gene

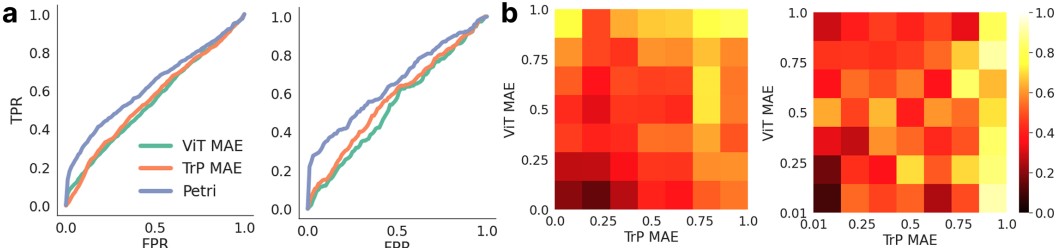

**Figure 3:** StringDB network edge metric analysis. The left panel of both sub-figures corresponds to the HepG2 dataset and the right to Perturb-Multi. **a**, ROC curves for PETRI and unimodal models (HepG2: PETRI AUC=0.628, ViT MAE AUC=0.549, TrP MAE AUC=0.556; Perturb-Multi: PETRI AUC=0.655, ViT MAE AUC=0.537, TrP MAE AUC=0.571). **b**, 2D histograms showing the pseudo classifier probabilities for unimodal models against the probabilities in PETRI.

embeddings. The similarities are treated as pseudo classifier probabilities and we evaluate the true positive rate (TPR) at 5% false positive rate (FPR) from the ROC curve. We expect this metric to be challenging because StringDB is neither cell nor phenotype specific and many single gene perturbations are expected to have weak or no effect.

Before aggregation for both metrics we robust center scale embeddings relative to a per replicate control and apply PCA and whitening without dimensionality reduction. Justification for and results without this preprocessing are in **A.5**. To form a multimodal perturbation profile, for the given perturbation we average modality-specific cell embeddings and then average across modalities.

## 4 EXPERIMENTS

We evaluate PETRI on two datasets and compare against CLIP and unimodal baselines.

**HepG2 (Matched Perturbations):** This dataset consists of matched OPS and Perturb-seq in HepG2 cells in four different chemical backgrounds. Cells were imaged with fluorescence and label-free microscopy, while a separate but matched population was profiled with whole transcriptome Flex sequencing. We included cells that received exactly one sgRNA from a CRISPR knockout library of 569, with four unique sgRNAs per target gene. Cells were grouped first by chemical background and then by sgRNA as in **Fig. 2b**; 16 cells per modality were sampled from each group to form a document (see **A.1**). 8 latent tokens were used per cell such that documents contained $16 \times 2 \times 8 = 256$ tokens (see **A.10**). The total dataset has 2M cells (0.9M images, 1.1M transcriptomics). We publicly release a gene-blinded version of this dataset together with this paper.

**Perturb-Multi (Matched Cells):** This dataset from Saunders et al. (2025) consists of spatial transcriptomics acquired from a single section of mouse liver tissue, including paired single-cell MERFISH measurements of 209 mRNAs and fluorescence images of 18 stained proteins. The dataset contains cells that received a sgRNA from a CRISPR knockout library of 203, with two unique sgRNAs per target gene. Cells were grouped by unique cell ID such that all groups contained a single cell; the set size was therefore 1 per modality and sampling was not required. Again, 8 latent tokens were used per cell, giving a document length of $1 \times 2 \times 8 = 16$ tokens.

To validate our hypothesis that joint reconstruction is robust even when the multimodal sets are not mutually informative, we additionally trained PETRI models on permuted data, where cells were randomly assigned to groups regardless of perturbation label.

Unless otherwise stated, PETRI cell embeddings were extracted directly from the output of the modality-specific encoders, i.e., before cell document creation and the MST. The MST's cross-modality attention encourages the upstream encoders to create tokens that are well aligned and compatible. As tokens approach the decoders, they become specialized for the specific reconstruction task of that training step. The information sharing that happens in the MST is a critical driver for multimodal integration, even if the layers themselves do not produce the best embeddings for downstream tasks. More generally, SSL methods commonly (Bordes et al., 2022) benefit from re-

**Table 1:** Evaluation of PETRI against unimodal and multimodal baselines (GC=Guide Consistency; StringDB=StringDB edge classification). CLIP* denotes two different models: using single cell pairs for Perturb-Multi and using mean aggregated perturbation-level pairs for HepG2 (see **A.3**). Values are median across three seeds for HepG2 and one seed for Perturb-multi.

| Modality | Method | Perturb-Multi | | HepG2 | |
|---|---|---|---|---|---|
| | | GC | StringDB | GC | StringDB |
| Transcriptomics | PCA | 0.030 | 0.073 | **0.304** | 0.150 |
| | scGPT - Pretrained | 0.059 | 0.078 | 0.048 | 0.057 |
| | TrP MAE | 0.059 | 0.109 | 0.195 | 0.144 |
| | PETRI Omics | - | - | 0.209 | 0.167 |
| Imaging | DINOv2 - Pretrained | 0.015 | 0.068 | 0.008 | 0.062 |
| | ViT MAE | 0.000 | 0.094 | 0.067 | 0.116 |
| | PETRI Image | - | - | 0.031 | 0.086 |
| Multimodal | TrP + ViT MAE Concat. | 0.000 | 0.099 | 0.155 | 0.153 |
| | TrP + ViT MAE Mean Cos. | 0.035 | 0.068 | 0.178 | 0.163 |
| | TrP + ViT MAE Max Cos. | 0.000 | 0.094 | 0.169 | 0.219 |
| | Cross-modal AE | - | - | 0.032 | 0.100 |
| | CLIP* | 0.000 | 0.057 | 0.051 | 0.174 |
| | PETRI Permuted Data | 0.163 | 0.260 | 0.274 | **0.255** |
| | PETRI | **0.208** | **0.260** | 0.278 | 0.242 |

moving layers. Crucially, this means that the jointly trained image and gene expression encoders can be independently run at inference time to embed cells in unmatched screening data as well.

## 4.1 PERTURBATION PROFILES FROM PETRI EMBEDDINGS RECAPITULATE KNOWN BIOLOGY

In this section, we benchmark PETRI for the task of aggregating multimodal data into a holistic perturbation embedding. As unimodal baselines, we tested strong pre-trained models, scGPT (Cui et al., 2024) and DINOv2 (Oquab et al., 2023), and modality-specific MAEs (TrP MAE and ViT MAE). For the HepG2 dataset only, we also experimented with variants of PETRI that operate on unimodal cell documents, which we designate PETRI Image and PETRI Omics. **Table 1** summarizes our findings. Compared to unimodal baselines, PETRI performs substantially better on both datasets and evaluation metrics. The only exception was PCA on gene expression for the HepG2 dataset, which showed slightly higher guide consistency but a much lower StringDB score. ROC curves for PETRI against unimodal MAEs show better detection of StringDB edges at all false positive rates (**Fig. 3a**).

Next, we considered simple late-fusion of unimodal perturbation profiles with two methods: (1) Mean or max aggregation of the modality-specific cosine similarity matrices and (2) concatenation of unimodal embeddings and computation of new similarity matrices. Max aggregation gave StringDB scores close to PETRI for the HepG2 dataset, though guide consistency showed no increase over TrP MAE. Cross-modal autoencoders (AE) provide an alternative late-fusion method, designed expressly for unpaired data like the HepG2 dataset. Using default hyperparameters for images and gene expression, we found that the latent space alignment enforced by the loss function prevented accurate reconstruction of RNA-seq counts. Results from this model were on par with unimodal image baselines, suggesting that gene expression was not effectively fused into the frozen image-only latent space. Additional late-fusion strategies are evaluated in **A.7**. PETRI uniformly outperformed all late-fusion approaches.

CLIP, our early-fusion baseline, performed worse than late-fusion, especially for Perturb-Multi. We included Perturb-Multi because we expected its cell-level pairs of protein-stained images and mRNA counts would be more appropriate for CLIP. We theorize that the two modalities may not have a close enough relationship for contrastive learning. As a test, we trained a ViT to regress the mRNA counts directly from protein images. On a held out validation set we found that 80% of mRNAs were predicted with $r^2 < 0.20$ (mean $r^2 = 0.117$; see **A.8**). To adapt CLIP for the HepG2 dataset, we used sets of perturbed cells, as in PETRI, and mean aggregated their profiles before computing the contrastive loss. This is effectively the architecture of CellCLIP without pre-trained and frozen encoders. Embeddings from this model performed worse than simple late-fusion baselines.

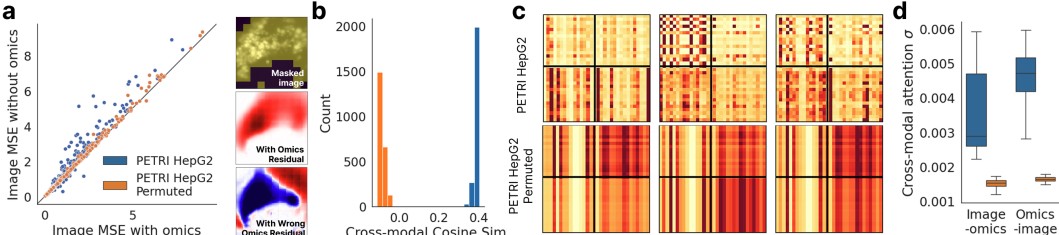

**Figure 4:** PETRI cross-modality information usage. **a**, Left: BODIPY reconstruction loss with vs. without access to perturbation-matched omics. Right: Example masked image of BODIPY channel (top) and residual maps between the image-only reconstruction and with matched vs. random omics (bottom two; red positive, blue negative). **b**, Cross-modality cosine similarities of treatment-aggregated embeddings (colors as in **a**). **c**, Representative attention heads; black lines separate image (top-left) and omics (bottom-right) token blocks; off-diagonals indicate cross-modal attention. **d**, Mean of standard deviations, $\sigma$, over rows in image-omics (upper right) and omics-image (lower left) attention map quadrants, from all attention heads in the MST layers (colors as in **a**).

To establish that PETRI incorporates information from both modalities in its aggregated perturbation profiles, we plotted the pseudo classifier probabilities against unimodal models (**Fig. 3b**). Probabilities for PETRI were highest when they were also high in both modalities. They remained high, but decreased, where the modalities disagreed. The correspondence is stronger for HepG2 than Perturb-Multi, which appears to favor gene expression over images. Overall, this shows that PETRI learns associations between gene pairs even when that association is only strongly visible in one modality.

A striking finding was that training PETRI on permuted variants of the datasets gave broadly similar results (worse on Perturb-Multi GC and better on HepG2 StringDB). Training on permuted data discourages cross-modality learning, but unimodal learning should not be affected; the model should just learn not to attend to the other modality. As such, the finding that PETRI is robust to permuted data is thus a positive result. Notably, it seems that by virtue of existing in the same space, unimodal PETRI embeddings can more effectively be aggregated into a multimodal profile with simple averaging than more sophisticated late-fusion methods on embeddings from separately trained models.

Importantly, note that the metrics evaluated here only consider image and gene expression embedding *averages*, not whether they are coherently integrated at a feature level. The following sections dig deeper and show that, with correctly matched data, PETRI actually learns cross-modality relationships with biological relevance that are useful beyond aggregate perturbation profiles.

## 4.2 PETRI INTERNALLY MAKES CROSS-MODALITY PREDICTIONS

To validate that PETRI uses cross-modality information, we performed an ablation analysis. We focused on the reconstruction of the BODIPY channel, which stains for lipid droplets, in the HepG2 dataset. This choice was motivated by our screening library that includes control perturbations known to regulate lipid droplet size, quantity, and distribution. Using a simple intensity threshold, we selectively masked out image patches containing droplets and evaluated the MSE loss for those patches. We calculated loss when cell documents only had latent tokens from the masked image and when additionally given access to latent tokens from a set of unmasked gene expression profiles.

Providing gene expression tokens from the same control perturbation decreased the *image* reconstruction loss for BODIPY relative to the image only case (**Fig. 4a**). The effect was sporadic across cells, but when present the MSE decrease was sizable—the same was not true for the model trained on permuted data, signaling the benefit of the document approach for cross-modal alignment. Inspecting an image where including gene expression had a large effect, we see an increase in predicted BODIPY intensity in the masked-out region over the image-only reconstruction. Providing gene expression from a different control perturbation known to *decrease* lipid droplets, we see a decrease in BODIPY intensity instead. In this targeted analysis, we conclusively showed that PETRI image reconstructions are influenced by transcriptomics data, and in biologically expected ways (see **A.11**).

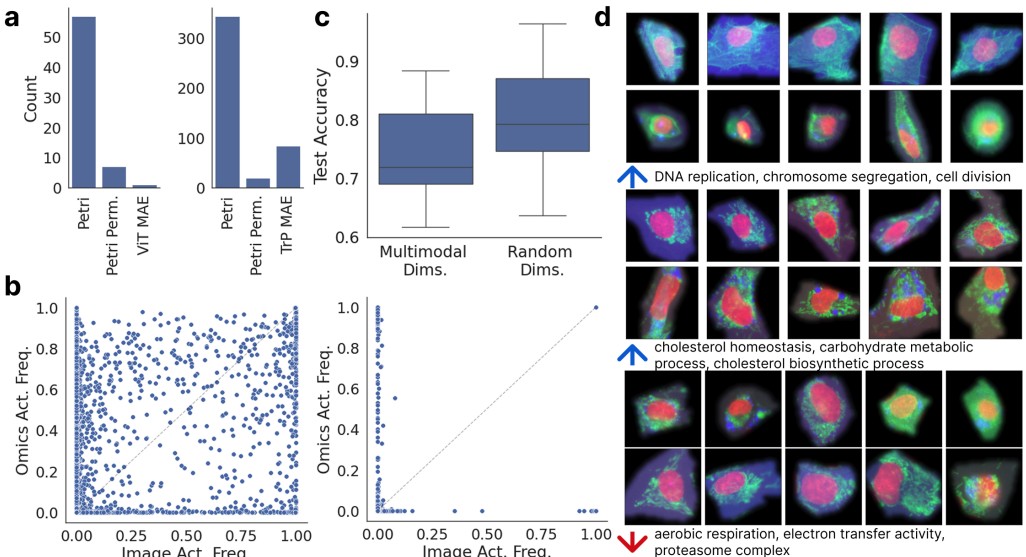

**Figure 5:** SAE analysis on HepG2. **a**, Number of perturbations with $\geq 1$ differentially activated SAE dimension in images (left) or omics (right). **b**, Fraction of cells activating each SAE dimension in images vs. omics (left) and on permuted data (right). **c**, Test-accuracies for classifiers trained to perform a batch–prediction task using multimodal SAE dimensions vs. randomly sampled dimensions (20 random draws per classifier). **d**, Representative images: top, non-activating samples; bottom, 99th-percentile activations. Arrows annotate the top three GO terms (pre-ranked enrichment; direction indicated). Channels: DAPI; BODIPY; Phalloidin (top); MitoProbe (bottom two).

Returning to the hypothesis that embeddings before the MST would be aligned across modalities, we measured the cosine similarity of unimodal perturbation profiles created from the PETRI modality-specific encoder outputs. Whereas the model trained on correctly matched data has a clear and positive cosine similarity, the model trained on permuted data shows near orthogonality (**Fig. 4b**). This orthogonality would not hurt performance on the evaluation metrics in the previous section, but does point to a bifurcation of the modalities in the latent space. Further evidence of this is suggested by visualizing the attention heads in the MST. Training on correctly matched data uniquely shows patterns of non-trivial and statistically significant ($p < 0.001$) cross-modality attention (**Fig. 4c,d**).

### 4.3 IDENTIFICATION OF MULTIMODAL CELLULAR PHENOTYPES

Having established that PETRI uses cross-modality information for reconstruction, we probed the concepts it learns to encode. To disentangle shared and modality-specific structure in PETRI embeddings, we trained BatchTop$K$ sparse autoencoders (SAEs) with 15,360 dimensions and $K = 500$.

We examined all SAE dimensions to test whether grouping cells by perturbation during training encouraged more salient perturbation-specific concepts than uninformed baselines. We performed a differential analysis of SAE activations for perturbations versus the negative control, controlling false discovery rate with Benjamini–Hochberg. PETRI had substantially more perturbations with at least one differentially activated dimension than unimodal models or the model trained on permuted data (**Fig. 5a**); this again supports that PETRI learns biologically relevant signal by coherently integrating modalities. In particular, cells within a document vary in orientation, intensity, and cell cycle, but a perturbation-enriched phenotype provides context that reduces uncertainty (e.g., the appropriate BODIPY intensity to predict), nudging the model toward concepts that distinguish perturbations. While supervised training also aligns concepts to perturbations, it implicitly assumes each perturbation has a unique and appreciable effect, which is often violated in CRISPR screens.

We then searched for concepts shared across modalities, defining "multimodal" dimensions as those activated in 10–90% of cells in both imaging *and* transcriptomics. PETRI produced 298 such dimen-

sions, compared with 0 for the permuted data model and 1 for CLIP (**Fig. 5b**); these results speak to PETRI's ability to align modalities in the latent space, resulting in multimodal concepts. Furthermore, if these concepts truly reflect biology common to both modalities, they should be less sensitive to modality-specific technical artifacts. We tested this by training logistic regression models to predict OPS well identity using either the 298 multimodal SAE dimensions or 298 randomly selected ones. Classifiers using the multimodal dimensions were significantly less accurate ($p < 0.001$; **Fig. 5c**), consistent with reduced encoding of undesirable well-specific technical factors.

Finally, we inspected the multimodal SAE dimensions for interpretability. For each dimension, we selected and compared images and transcriptomic profiles with zero activation against those with activation above the 99th percentile and asked whether there were statistically significant differences between these sets of images and transcriptomic profiles, respectively. For images, we ran differential analysis on handcrafted features of fluorescence intensity and nuclear and lipid droplet morphology. For transcriptomics, we performed differential expression analysis and computed pre-rank enrichment against Gene Ontology (GO) terms. In total, 127 of the 298 dimensions showed significant differences for at least one image feature and enrichment in at least one GO term.

To further interrogate these multimodal PETRI SAE dimensions for relevant biology, we searched by keywords for SAE dimensions that showed enrichment in terms related to phenotypes we know should be present in our dataset: cell cycle, lipid metabolism, and mitochondrial activity. Corresponding images for those dimensions revealed interpretable biological concepts including DNA replication (correlating with nucleus shape and DAPI intensity), cholesterol homeostasis (correlating with lipid droplet size and quantity), and aerobic respiration (correlating with mitochondrial fusion and network structure) (**Fig. 5d**).

These results demonstrate that PETRI can be leveraged to link molecular states to morphological phenotypes at single-cell resolution. This enables researchers to confidently prioritize morphological changes that are validated by a corresponding molecular signature, ensuring they are both real and biologically relevant. SAEs trained on unimodal embeddings cannot be used for this purpose without first finding correspondences between separately learned concepts.

## 5 CONCLUSION

We presented PETRI, an early fusion approach for learning multimodal single cell embeddings from unpaired image and transcriptomic data; the approach is robust to the presence of only weak or variable cross-modal alignment. We demonstrated that PETRI-learned perturbation profiles (1) recover gene relationships better than unimodal and late-fusion baselines, and (2) encode batch-robust, biologically meaningful concepts that can be disentangled with sparse autoencoders.

A key insight is that joint reconstruction over context-grouped data can induce meaningful multimodal alignment without explicit cross-modal losses. PETRI demonstrates this, opening a path to unifying historically separate screening modalities. At the same time, our results highlight limits of prevailing proxies (e.g., guide consistency, protein–protein interaction prediction): useful as benchmarks, but insufficient to capture the biologically meaningful structure revealed by our downstream analyses. Instead, there is a need for task-grounded evaluation frameworks tailored to multimodal phenotypic screening and therapeutic discovery, with metrics that directly assess biological utility.

Several other important questions remain: How closely matched must experimental contexts be to enable cross-modality learning? Is it possible to incorporate other biological priors? For instance, instead of documents that focus on a single perturbation, what if we centered them on protein complexes or pathways? Our framework provides a foundation for exploring these questions and suggests that strategic contextual grouping could serve as a mechanism for easily incorporating domain knowledge into representation learning.

Although PETRI is designed for images and gene expression, the core idea of using joint reconstruction from context-aligned cellular documents could be adapted to other modalities. As diverse omics technologies continue to proliferate, approaches that can integrate complementary views of cellular state without requiring perfect experimental alignment will become increasingly valuable for advancing our understanding of cellular biology and accelerating therapeutic discovery.

REPRODUCIBILITY STATEMENT

Code and the blinded HepG2 dataset will be made public. All results aside from StringDB metrics on the HepG2 dataset will be reproducible from the provided code and data.

ACKNOWLEDGEMENTS

We thank members of the Data Science and Machine Learning department at Insitro for providing early feedback on the results presented in this work.

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

# A   APPENDIX

## A.1   HEPG2 DATASET

Briefly, a pool of constitutive Cas9-expressing hepatocellular carcinoma cells (HepG2s) received approximately one sgRNA per cell from a CRISPR knockout library of 569 genes, with four unique sgRNAs per target gene. Cells were expanded as a uniform pool to maximize parity between imaging and transcriptomic populations, then treated with one of four chemical treatments. Cells were imaged with fluorescence at 20X magnification by staining the cells with a variation of CellPaint, which included HOESCHT, PHALLOIDIN, BODIPY, and MITOPROBE to label DNA, actin, neutral lipids, and mitochondrial rRNA, respectively. Cells were additionally imaged via quantitative phase imaging and brightfield. Imaging was followed by sgRNA amplification and in-situ sequencing. A parallel population was used for the creation of the transcriptomic dataset, which was generated using Flex technology, with probes reading out both the transcription, as well as the gRNA sequences, within each cell.

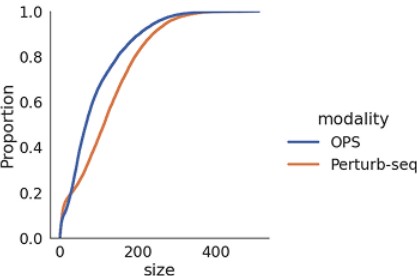

**Figure S1:** ECDF of cell counts per perturbation group (sgRNA)

Due to fitness effects, the number of barcoded cells that received each sgRNA varies considerably **(Figure S1)**. 19% of perturbation groups had fewer than 16 cells in one or the other modality, which motivated the choice of 16 as the default set size for PETRI. With larger set sizes and sampling with replacement, the possibility of leakage increases, e.g., the same cell with different masking patterns can appear in the same set.

## A.2   PETRI IMPLEMENTATION

**Unimodal encoders and Multimodal Set Transformer (MST)**   The image encoder is a standard ViT-Base model (Dosovitskiy et al., 2020) with 85M parameters. The gene expression encoder is a Perceiver with four cross-attention layers and 12 self-attention layers, where each cross-attention layer is followed by three self-attention layers. The total parameter count is 131M, of which 16M correspond to learnable gene embeddings. To match ViT-Base, the embedding dimension is 768. After the unimodal encoders, the MST is a four layer transformer comprised of ViT-style self-attention blocks with a total of 14M parameters.

**Decoders**   The image decoder architecture follows Masked Autoencoders and using a ViT with embedding dimension of 512 and eight transformer layers amounting to 26M parameters. The gene expression decoder is a simple three-layer MLP with hidden dimension of 128 and final output dimension of either 18082 for HepG2 or 209 for Perturb-Multi. It consists of 2M parameters.

The total parameter count is 259M with roughly equal numbers of parameters for each modality. However, the gene expression-related parameters are almost entirely in the encoder whereas a substantial fraction of the image-related parameters are in the decoder.

## A.3   CLIP BASELINE IMPLEMENTATION

To adapt CLIP to our setting, we used the same architecture as PETRI but removed the decoders and Multimodal Set Transformer. For Perturb-Multi, with cell pairs, we used a batch size of 4096 and the standard CLIP loss over positive and negative pairs. For HepG2, with perturbation-level

matching, we used sets of 16 cells sampled from the same perturbation as positive pairs. Before computing the loss, we mean-aggregated the embeddings of all cells in the set. The batch size was 4096 cells per modality; however, after aggregation there were $4096 \div 16 = 256$ pairs over which to compute the CLIP loss.

## A.4 MODEL TRAINING

PETRI and baseline models were all trained for 100 epochs. For the HepG2 dataset, every epoch included exactly 30 samples of cells drawn from each perturbation group. This balanced sampling was consistent for all models that we trained, regardless of whether they required sets of cells or not. For Perturb-Multi, with cell-level pairs, this sampling was unnecessary and an epoch included each cell just once. In total, HepG2 models trained for 250K iterations and Perturb-Multi models trained for 145K iterations. Training took 2 days on a single node with 8 H100 GPUs and a batch size of 1536 masked images and/or expression profiles.

**Image augmentations and patching**    Perturb-Multi and HepG2 images were augmented with random vertical and horizontal flips, 360 degree rotations and 5% translations. To account for cell sizes, we used center crops of 64 pixels for Perturb-Multi and 80 pixels for HepG2. ViT patch size was 8 pixels for both.

**Loss weighting**    For the HepG2 dataset, we used a weight of 1.0 for the image MSE and $1 \times 10^{-4}$ for the negative binomial loss. For Perturb-Multi, we used a weight of 1.0 for the image MSE and 0.01 for the mRNA count MSE.

## A.5 EMBEDDING POSTPROCESSING AND EVALUATION

We preprocessed single cell embeddings before aggregating them into perturbation profiles and computing guide consistency and StringDB network edge metrics. For HepG2 image embeddings, we performed robust center scaling (RCC), i.e., we subtracted the median embedding of the intergenic control cells in each OPS well and divided by the interquartile range. Standardizing by per replicate controls is a common method for reducing batch effects introduced by natural well-to-well variability in culture media or fluorescence intensities and empirically works well for ViT embeddings (Kraus et al., 2024). For HepG2 gene expression embeddings, we performed RCC using the global intergenic controls. Cells from the four chemical backgrounds were preprocessed independently and the evaluation metrics for each background were averaged.

Since Perturb-Multi cells all come from a single batch, we used global statistics from the non-targeting control. PCA and whitening promote isotropy and enhance semantic search for language embeddings (Diera et al., 2024; Sasaki et al., 2023), with similar benefits for perturbation embeddings (Kraus et al., 2024).

**Table S1** shows that StringDB metric evaluation on the raw embeddings gives no better than random performance (0.05) on HepG2.

**Table S1:** Evaluation of unimodal and multimodal embeddings without preprocessing on the HepG2 dataset for the StringDB edge classification metric.

| Modality | Method | HepG2 StringDB |
|---|---|---|
| Transcriptomics | PCA | 0.028 |
| | scGPT - Pretrained | 0.043 |
| | TrP MAE | 0.037 |
| | PETRI Omics | 0.040 |
| Imaging | DINOv2 - Pretrained | 0.050 |
| | ViT MAE | 0.040 |
| | PETRI Image | 0.043 |
| Multimodal | CLIP* | 0.040 |
| | PETRI Permuted Data | 0.045 |
| | PETRI | 0.031 |

## A.6 INFERENCE PROCEDURE AND INTERMEDIATE LAYER EMBEDDINGS

We chose to use PETRI embeddings from the outputs of the unimodal encoders because they can be computed without access to multimodal groups of cells. However, we also considered using cell embeddings from the intermediate layers of the Multimodal Set Transformer. This required a more complex inference procedure.

**Deterministic multimodal set inference.** Each perturbation group was first shuffled to mitigate potential batch effects, then wrap-around padded to ensure the total number of cells in the group was divisible by the set size. The padded indices were then partitioned into non-overlapping sets of fixed size, creating a deterministic enumeration. The embeddings for duplicated cells were averaged such that the final inference result had no duplicates. When the number of available sets differed between modalities for the same perturbation, the modality with fewer sets was cyclically repeated to match the longer modality. This guarantees that every cell appears at least once in each modality while maintaining proper multimodal alignment between sets.

**Table S2:** Evaluation of PETRI embeddings from intermediate layers on the HepG2 dataset (GC=Guide Consistency; StringDB=StringDB edge classification).

| Layer | HepG2 | |
|---|---|---|
| | GC | StringDB |
| Before MST (default) | **0.260** | **0.242** |
| After MST Layer 1 | 0.202 | 0.200 |
| After MST Layer 2 | 0.211 | 0.197 |
| After MST Layer 3 | 0.220 | 0.190 |
| After MST Layer 4 | 0.221 | 0.186 |

**Table S2** summarizes the results and shows that the embeddings taken directly from the unimodal encoders give the best guide consistency and StringDB scores on the HepG2 dataset.

## A.7 LATE-FUSION METHODS

We tested three late-fusion methods, all of which make use of aggregated embeddings from ViT MAE and TrP MAE after RCC and whitening:

1. **Concatenation:** Imaging and transcriptomics each had 768 dimension embeddings, which we concatenated to produce 1,536 dimensions. With these we computed pairwise cosine similarities between target gene or sgRNA profiles, depending on the metric.

2. **CCA:** We fit CCA with 30 components to find a shared space between embeddings from the two modalities. Embeddings from both modalities were projected into this space and we either concatenated or averaged them directly.

3. **Cosine similarity matrix aggregation:** Instead of working with the unimodal embeddings, we directly aggregated the cosine similarity matrices from each modality by min, max, mean, or median.

**Table S3** summarizes the results.

## A.8 PERTURB-MULTI ViT REGRESSION

We trained a ViT-Base with the same settings as those in **A.4**. The class token was processed with a two-layer MLP to regress the 209 normalized mRNA counts in Perturb-Multi, loss was evaluated with MSE. **Fig. S2** shows the distribution of $r^2$ values for the mRNAs evaluated on a randomly chosen held out test set containing 20% of total cells.

## A.9 PERMUTED DATASET EVALUATION

We observed that the performance on guide consistency and StringDB network edge metrics was roughly equivalent between PETRI models trained on the correctly grouped and matched versions of

**Table S3:** TrP + ViT MAE late-fusion results (GC=Guide Consistency; StringDB=StringDB edge classification).

| Method | Perturb-Multi | | HepG2 | |
|---|---|---|---|---|
| | GC | StringDB | GC | StringDB |
| TrP + ViT MAE Concat. | 0.000 | 0.099 | 0.155 | 0.153 |
| TrP + ViT MAE CCA Concat. | 0.000 | **0.104** | 0.019 | 0.079 |
| TrP + ViT MAE CCA Mean. | 0.000 | 0.099 | 0.020 | 0.079 |
| TrP + ViT MAE Min Cos. | 0.000 | 0.094 | 0.115 | 0.116 |
| TrP + ViT MAE Mean Cos. | 0.035 | 0.068 | **0.178** | 0.163 |
| TrP + ViT MAE Median Cos. | 0.035 | 0.068 | 0.178 | 0.163 |
| TrP + ViT MAE Max Cos. | **0.040** | 0.094 | 0.169 | **0.219** |

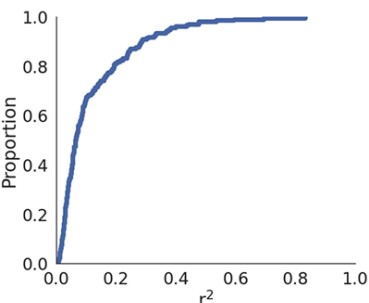

**Figure S2:** ECDF of $r^2$ scores for predictions on each of the 209 mRNA counts in Perturb-Multi.

the benchmark datasets versus their permuted versions. **Figure S3** shows the ROC curves and Venn diagrams for StringDB metrics. The results suggest that training on correct or permuted datasets leads to identification of broadly similar gene pairs.

### A.10 DOCUMENT LENGTH AND LATENT TOKEN COUNT

The resampling mechanism allows us to trade-off the number of latent tokens for the number of cells in a document. In the main experiments, we used 8 tokens per cell and a set size of 16. At least for the available evaluation metrics, we did not observe a clear pattern of improvement or degradation from adjusting these parameters (**Figure S4**) and absolute differences were small. With smaller set sizes we might expect to see less cross-modality learning, especially if perturbations induce heterogeneous responses.

### A.11 RECONSTRUCTION LOSS ABLATION ANALYSIS

**Fig. 4a** shows the effect of a targeted ablation analysis on image reconstruction loss. At the aggregate level, we did not notice clear differences in reconstruction loss for images or gene expression in the HepG2 dataset when removing access to the other modality. However, we do observe that image reconstruction loss is lower overall than the model trained on permuted data **Figure S5a,b**.

For Perturb-Multi, on the other hand, there is a statistically significant ($p \ll 0.05$) increase in loss when access to the other modality is removed. This effect disappears for the model trained on permuted data. Again, supporting the notion that the model learns to ignore irrelevant multimodal information.

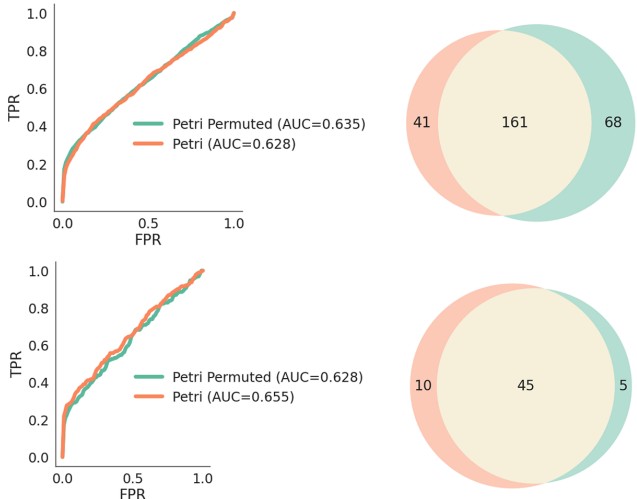

**Figure S3:** ROC curves and Venn diagrams for TP StringDB gene pair detections. **Top:** HepG2. **Bottom:** Perturb-Multi

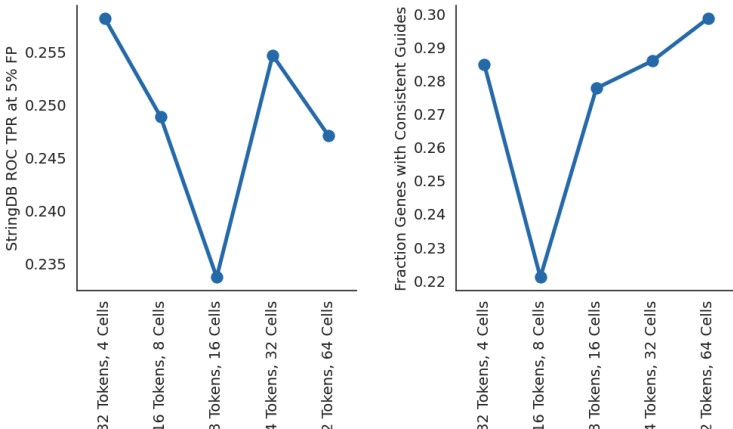

**Figure S4: Left:** StringDB metric on HepG2 with different combinations of latent tokens and set sizes. The listed number of cells is per modality. **Right:** Same for the guide consistency metric.

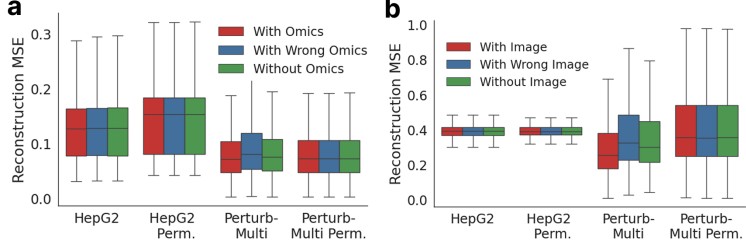

**Figure S5: a.** Image reconstruction loss across all channels when randomly masking 75% of patches; includes no access to omics, access to correctly paired omics, and access to random omics. **b.** Average reconstruction loss across all genes when randomly masking 75% of gene tokens; includes no access to images, access to correctly paired images, and access to random images.

