# OpenReview forum: "PETRI: Learning Unified Cell Embeddings from Unpaired Modalities via Early-Fusion Joint Reconstruction"
_ICLR.cc/2026/Conference — ICLR 2026 Poster_

### Official Review · Reviewer_Gmu9 · 2025-10-20

**Soundness:** 3
**Presentation:** 3
**Contribution:** 2
**Rating:** 2
**Confidence:** 5

**Summary:**

The paper introduces PETRI, an early-fusion transformer framework designed to learn unified single-cell embeddings from unpaired cell morphology imaging and transcriptomic data. PETRI groups cells sharing the same perturbation into multimodal “documents” and performs masked joint reconstruction across modalities, enabling it to share information between imaging and gene expression while preserving modality-specific signals. Evaluated on both HepG2 (unpaired OPS and Perturb-seq) and Perturb-Multi (paired spatial transcriptomics) datasets, PETRI outperforms contrastive and unimodal baselines in recovering biologically meaningful perturbation profiles and cross-modal relationships. Analyses with sparse autoencoders further show that PETRI discovers interpretable multimodal cellular concepts linking morphological and molecular phenotypes, demonstrating a scalable approach for integrating heterogeneous biological assays without requiring cell-level pairing.

**Strengths:**

- PETRI focuses on integrating unpaired cellular modalities (images + transcriptomics), a setting that is much more realistic in biological experiments where cell-level pairing is infeasible.

**Weaknesses:**

- The paper motivates the need for integrating unpaired multimodal data but does not clearly articulate why existing late-fusion or generative alignment methods (e.g., cross-modal autoencoders, cross-domain VAEs) are insufficient.

- Technical contribution is somewhat limited. The major novelty lies on the way to construct paired cell image-gene expression data from unpaired data (i.e., documents). Other technical parts — ViT/MAE for imaging, perceiver for transcriptomics, and multimodal masked modeling [1] are not novel idea in multimodal learning.

- The choice of early fusion via joint reconstruction is not analytically or empirically contrasted with late-fusion or contrastive paradigms. It is unclear under what conditions early-fusion offers superior representational alignment or when it could fail (e.g., modalities with minimal mutual information).

- The two core metrics—Guide Consistency and StringDB edge classification—capture perturbation-level similarity but do not fully reflect biological utility or multimodal alignment quality. There is no quantitative measure of cross-modal transfer or downstream biological inference.

- Comparisons are restricted to unimodal MAEs, CLIP, and a few pre-trained models. Recent multimodal generative or diffusion-based frameworks (e.g., MoCa, scMultiVI, CellCLIP) are discussed but not empirically included, limiting fairness and completeness of the comparison.

- The experiments lacks statistical rigor and scalability analysis. The authors did not provide error bars, confidence intervals, or multiple runs. Scalability to larger or more heterogeneous datasets, and sensitivity to document size, masking ratio, or latent dimension, are not thoroughly studied.

- The study stops at embedding analysis; there is no demonstration that PETRI improves biological discovery tasks such as hit identification, functional gene grouping, or mechanism-of-action prediction.

References:

[1] Bachmann, Roman, et al. "Multimae: Multi-modal multi-task masked autoencoders." European Conference on Computer Vision. Cham: Springer Nature Switzerland, 2022.

**Questions:**

- What specific limitations of existing contrastive or late-fusion multimodal methods (e.g., CLIP, MultiVI, CellCLIP) does PETRI aim to overcome? Can you provide conceptual or empirical evidence that early-fusion leads to superior integration under unpaired settings, rather than simply matching performance?

- Beyond improving embedding quality, what concrete biological or biomedical applications does PETRI enable? Could the authors provide one clear use case where PETRI’s unified embeddings yield novel biological insight that unimodal or late-fusion methods cannot?

- Why is reconstruction-based self-supervision preferable to contrastive or generative alignment (e.g., VAE-style, diffusion-based)? Did you observe any stability issues or mode collapse when using the joint reconstruction objective?

- The paper claims PETRI learns meaningful cross-modal attention patterns. Can you quantify or statistically validate this, rather than relying solely on visual inspection? Could you measure how much of the reconstruction or embedding variance is attributable to cross-modal information versus unimodal content?

-  Are the improvements over baselines statistically significant (e.g., via multiple random seeds or bootstrapping)? Can the authors report variance or confidence intervals to confirm the robustness of observed gains?

---

> ### Author Response · Authors · 2025-12-03
> **Response to reviewer Gmu9 - Baselines and applications**
>
> We thank you for recognizing that PETRI addresses a critical and realistic challenge in biological experimentation: integrating unpaired cellular modalities. We appreciate the thoughtful critique regarding our baselines, novelty, and evaluation metrics. We address these points below.
>
> **On the suitability of Late-Fusion and Generative Alignment**
>
> The limitation of models like MultiVI is their reliance on cell-level pairing. MoCa uses existing documents of natural language and images and is not designed for biological modalities. Our implementation of CLIP for the HepG2 dataset is effectively the same as CellCLIP, but with early-fusion. CellCLIP itself is for images and natural language as well. We have clarified these distinctions in the text to ensure the fairness of our comparison is understood.
>
> As an additional baseline, we have included cross-modal autoencoders as a late-fusion baseline (it learns the latent space for images first and then trains a transcriptomics model that aligns to the image space). We observed significant training instability with different seeds; even models that trained successfully struggled to reconstruct transcriptomics data accurately given the hard to satisfy constraint that its embedding space align with image space. Performance of cross-modal autoencoders was similar to image-only baselines on our evaluation metrics (Table 1). In contrast, we have not experienced training instability with PETRI models.
>
> **Technical Novelty and Architecture**
>
> Regarding your comment on technical novelty, we acknowledge that PETRI utilizes established components (ViT, Perceiver) and actually view that as a strength of our approach, demonstrating that our method of context-based integration is robust without requiring custom architectures. The core innovation of PETRI is not the components themselves, but the integration strategy: specifically, the "document" construction combined with aggressive token resampling. Our work demonstrates that we can achieve computationally tractable and effective cross-modal learning even with aggressive token resampling and weakly overlapping data, a finding that is non-trivial. This architecture solves the specific logistical problem of how to fuse sets of unpaired, high-dimensional biological data—a problem standard multimodal masked modeling does not address out-of-the-box. Additionally, our proposed approach for early fusion provides a plug-and-play framework for considering alternative modality-specific encoders.
>
> **Early-Fusion vs. Late-Fusion and Contrastive Baselines**
>
> We emphasize that we did compare PETRI against late-fusion and contrastive baselines in **Table 1** and **Table S3**. Specifically, "TrP + ViT MAE Concat" (late fusion) and our adapted CLIP implementation (contrastive) both underperformed PETRI on Guide Consistency and StringDB metrics. Contrastive learning (CLIP/CellCLIP) inherently relies on strong positive pairs; in both the HepG2 dataset with unpaired cells and limited unique treatments (~2,200) as well as the cell-paired Perturb-multi dataset we demonstrated that weak pairs renders contrastive learning ineffective. PETRI’s joint reconstruction objective offers a mechanism to discover these weak signals without assuming their existence.
>
> **Biological Utility and Evaluation Metrics**
>
> We appreciate the feedback regarding the biological utility of Guide Consistency and StringDB classification. While these are proxies, they are standard benchmarks for representation learning in this domain. Cross-modal transfer or nearest neighbor retrieval are also common metrics; however, they are unhelpful when modalities do not have substantial mutual information. Mechanism-of-action prediction would require chemical screening data, but only genetic screening data was available. Lastly, in this space, it is common to present methods development work separately from major scientific discoveries like hit identification.
> We believe that the results on StringDB sufficiently demonstrate the value of PETRI for functional gene grouping – no existing early- or late-fusion methods that we tested were as effective at integrating the information from both modalities into informative perturbation-level profiles. The work on SAEs, while not conclusive, also points to applications for batch correction and hypothesis generation, both of which are very important tasks and were underemphasized in the original submission.

---

> ### Author Response · Authors · 2025-12-03
> **Response to reviewer Gmu9 - Additional quantitative evaluation**
>
> **Statistical Rigor and Baselines**
>
> We acknowledge the need for greater statistical rigor. We have updated our results with multiple training runs with different random seeds (N=3) for the HepG2 dataset. Larger and more heterogeneous datasets do not exist for testing methods like PETRI, this motivates the release of a new dataset in this work. Sensitivity to document size and number of latent tokens per cell are reported in **Figure S4**.
>
> **Cross-Modal Attention**
>
> Finally, regarding the validation of cross-modal attention, we have added a quantitative analysis. We calculated the mean of standard deviations of cross-modal attention weights across cells, an approximate measure of how specific. We found that the attention PETRI pays to the other modality is statistically significantly higher (p < 0.001) compared to a baseline trained on permuted (unmatched) data. This confirms that the cross-modal dependencies we visualized are a systematic property of the model, not just anecdotes.

---

### Official Review · Reviewer_WChQ · 2025-10-29

**Soundness:** 3
**Presentation:** 4
**Contribution:** 4
**Rating:** 8
**Confidence:** 4

**Summary:**

This work presents PETRI, a multimodal transformer for biological data that unifies cell-level transcriptional gene-expression readouts with morphology images. PETRI is a self-supervised regime that combines two masked autoencoders for each data modality and uses latent token resampling to deal with the difficulties of very long sequence lengths caused by gene expression data. The model is demonstrated to perform quite well over two different datasets in terms of guide perturbation consistency and stringdb relationship retrieval against a comprehensive suit of baselines (such as unimodal methods, multimodal baselines like simple embedding concatenation and CLIP). The authors furthermore apply sparse autoencoders to analyze the multimodal concepts learned by the model, with an interesting case-study on the GO-term enrichment of these dimensions and the interpretable biological concepts obtained (e.g. DNA replication).

**Strengths:**

- Releasing a new multimodal public dataset is highly valuable.
- Valuable multimodal architecture: the PETRI architecture uses creative means to process replicates of the same perturbation across the two modalities, and the use of latent tokens addresses difficulties surrounding too many tokens that would otherwise be present.
- Clear and high quality results: Robust benchmarking analysis with appropriate baselines is pursued by comparing their regime to unimodal and late-fusion multimodal models, and the performance improvements of PETRI hold.
- Original and very significant: Highly relevant task is pursued and comparing results are obtained, indicating the important complementarity of multimodal biological screening assays in providing more comprehensive understandings of perturbations when unified together in the context of early-fusion, a highly motivating result that many may have held intuitively but, prior to this work, had yet to be thoroughly broached.
- Creative additional investigation into the nature of their multimodal representation learning model: SAE is used to analyze their multimodal model demonstrates biologically relevant compelling and interesting results (see Figure 5)
- Many avenues for relevant future work and additional research are opened up as a result of this work. I can think of a multitude; for example: (1) multidataset training: it could be interesting, e.g., to train a combined model on both Perturb-Multi and HEPG2; (2) exploring finetuning, e.g. it could make sense to pretrain a generic microscopy MAE on JUMP-CP data and then finetune that architecture here in PETRI, rather than training the image MAE from scratch; (3) varying the mask ratio, sometimes entirely masking the entire other modality, as being able to predict the other modality from one (e.g. if the transcriptomics is masked 100% and morphology only 50%) could be quite valuable if it works.

**Weaknesses:**

The only weakness I see in this paper is secondary, namely that it is lacking situated references to various related work that has pursued similar studies. For some examples, I think it could be fruitful to mention the following highly related papers to this work:
- Simple methods like control or mean baselines turn out to be hard-to-beat in transcriptomics and could be worth discussing in the context of their benchmarking; a variety of papers discuss this, e.g.: Ahlmann-Eltze, Constantin, Wolfgang Huber, and Simon Anders. "Deep-learning-based gene perturbation effect prediction does not yet outperform simple linear baselines." Nature Methods (2025): 1-5.
- The PCA with whitening approach is quite important embedding post-processing step and discussed in more detail here for both transcriptomics and morphology: Celik, Safiye, et al. "Building, benchmarking, and exploring perturbative maps of transcriptional and morphological data." PLOS Computational Biology 20.10 (2024): e1012463.
- For the latest in high-performing microscopy MAEs: Pham, Chau, Juan C. Caicedo, and Bryan A. Plummer. "ChA-MAEViT: Unifying Channel-Aware Masked Autoencoders and Multi-Channel Vision Transformers for Improved Cross-Channel Learning." arXiv preprint arXiv:2503.19331 (2025), NeurIPS 2025.
- The Kraus et al. citation in this work points to the original workshop paper, which was actually expanded upon in the following archival paper accepted at CVPR. Furthermore, it may be worth considering incorporating the improvement applied to the MAE loss function (i.e., the fourier-transform reconstruction loss) to improve the quality of the morphology MAEs trained in this work: Kraus, Oren, et al. "Masked autoencoders for microscopy are scalable learners of cellular biology." Proceedings of the IEEE/CVF Conference on Computer Vision and Pattern Recognition, CVPR 2024.
- On the general topic of multimodal unification of transcriptomics data with morphological features, this paper seems particularly relevant: Bendidi, Ihab, et al. "A Cross Modal Knowledge Distillation & Data Augmentation Recipe for Improving Transcriptomics Representations through Morphological Features." arXiv preprint arXiv:2505.21317 (2025), ICML 2025.
- Pretrained Dino-v2 has been evaluated on microscopy data in the following work, which also discusses further advancements to MAEs in the context of microscopy (e.g. the importance of curated training data and especially crucial is linear probing to identify intermediate layers of the ViT which obtain higher guide consistency and relationship prediction), which could be mentioned with the respect to e.g. the intermediate layer MST analysis in section A.6: Kenyon-Dean, Kian, et al. "Vitally consistent: Scaling biological representation learning for cell microscopy." arXiv preprint arXiv:2411.02572 (2024), ICML 2025. (This work also publicly released of a CA-MAE ViT-S on HuggingFace, which is worth considering for baseline comparison: https://huggingface.co/recursionpharma/OpenPhenom)
- On the topic of applying SAEs to biological foundation models, see: Donhauser, Konstantin, et al. "Towards scientific discovery with dictionary learning: Extracting biological concepts from microscopy foundation models." arXiv preprint arXiv:2412.16247 (2024), ICML 2025.

**Questions:**

What does it mean that the dataset you will release will be "blinded"? Would others be able to reproduce these results from the public dataset release, or are perturbation labels required to do so?

While described as *unpaired* data here, is the context here not more typically described as *weakly paired*, i.e. same perturbation but different cells? It seems that you are sampling groups of samples with matching guide perturbation labels from the two modalities -- i.e. line 154 you describe that you "Create batches where cells are grouped by perturbation to form sets."

---

> ### Author Response · Authors · 2025-12-03
> **Response to reviewer WChQ**
>
> We thank you for your enthusiastic support and for recognizing the high value of our dataset, architecture, and results. We are particularly gratified that you found the "multimodal document" approach creative and the Sparse Autoencoder (SAE) analysis compelling. We also appreciate the excellent list of related works and future directions. Below, we address your specific suggestions and questions.
>
> **Related Work and Citations**
>
> We agree that the manuscript benefits significantly from situating PETRI within the broader context of recent advances in microscopy and transcriptomics. We have incorporated all the suggested references into the revised manuscript. Specifically:
> - **MAE Improvements**: We have added citations for ChA-MAEViT (Pham et al.) and the updated Kraus et al. CVPR paper.
> -**Cross-Modal Integration**: We should note that while Bendidi et al. (Cross Modal Knowledge Distillation) is relevant, it uses bulk instead of single cell data and focuses on enriching unimodal embeddings.
>
> **Clarification on "Blinded" Dataset and Reproducibility**
>
> You asked what "blinded" entails and how it affects reproducibility. "Blinded" means the specific identities of the perturbations (e.g., Gene X, Compound Y) are anonymized, but the structure of the data (e.g., Perturbation A in images corresponds to Perturbation A in transcriptomics) is preserved. All results involving model training, architectural benchmarking, and stability analysis are fully reproducible with the released code and data. The only results that cannot be externally reproduced without the unblinded key are the specific biological validations requiring external database lookups (e.g., StringDB edge recovery), as these require knowing the gene names.
>
> **Terminology: "Unpaired" vs. "Matched"**
>
> We agree that “weakly paired” or “matched” could also be appropriate terms. Our specific setting is perhaps best described as: "Unpaired cells from matched perturbation contexts." “Weakly paired” is also valid but might be confused with “weakly labeled”, another common problem when working with single cell data. For simplicity and to align with existing literature, we will continue to use "unpaired" to denote *different cells* and reserve the term “paired” as strictly meaning *same cells*.
>
> **Masking Ratios**
>
> Regarding your suggestion to vary mask ratios (e.g., masking 100% of one modality) we effectively performed a local version of this in our "BODIPY ablation" (Figure 4a): we masked almost 100% of the lipid droplet information in the image channel to see if the model could recover it solely from the transcriptomic context. Extending this to full-modality masking is an excellent suggestion for possible applications of PETRI to cross-modality imputation.

---

### Official Review · Reviewer_DpdC · 2025-10-30

**Soundness:** 2
**Presentation:** 3
**Contribution:** 3
**Rating:** 4
**Confidence:** 2

**Summary:**

This paper presents PETRI, a early-fusion transformer framework designed to learn unified cell embeddings from unpaired single-cell imaging and transcriptomics data. The core idea is to group cells by a shared experimental context (e.g., a perturbation) into "multimodal documents" and train the model on a masked joint reconstruction task, enabling information sharing via cross-modal attention. To make this document-based approach computationally tractable, the authors employ an aggressive token resampling strategy. Through experiments on two datasets, the authors demonstrate that PETRI's unified embeddings outperform unimodal and late-fusion approaches on biological benchmarks like Guide Consistency and StringDB edge classification. Furthermore, they show that applying sparse autoencoders to the embeddings reveals biologically interpretable, multimodal concepts that are robust to technical confounders, and they validate that the model indeed uses cross-modal information for reconstruction. To facilitate further research, the authors also introduce and release a new large-scale dataset.

**Strengths:**

1. The paper proposes PETRI, a framework that effectively integrates unpaired imaging and transcriptomics data via a joint reconstruction task, addressing a challenge in the field.
2. The work provides strong evidence of model interpretability through multiple analyses. It demonstrates cross-modal information transfer via targeted ablation studies and attention matrix visualization. Applying sparse autoencoders (SAEs) reveals biologically meaningful, multimodal concepts that are shown to be robust to technical confounders.
3. The authors contribute a valuable new large-scale dataset of matched imaging and omics data.

**Weaknesses:**

1. The paper's best results come from embeddings taken before the data passes through the Multimodal Set Transformer (MST), the core component designed for fusion. The appendix (Table S2) confirms that performance on key benchmarks actually gets worse after the MST. This is counterintuitive and questions the practical benefit of the MST for the paper's main evaluation tasks.
2. The analyses showing that the model learns biological concepts (Sections 4.2 and 4.3) rely heavily on specific, illustrative examples (e.g., reconstructing the BODIPY channel, showing a few representative images). The paper lacks broad, quantitative metrics to prove that these impressive findings are consistent across the entire dataset and not just in a few hand-picked cases.
3. The comparison against the CLIP model may be unfair. CLIP is designed to work with perfectly paired data, but the main dataset in this paper is unpaired. The authors had to adapt CLIP for this different problem, which likely means it could not perform at its best. This may exaggerate the performance advantage of PETRI.
4. The PETRI model is very computationally expensive, requiring powerful and specialized hardware to run. This is because it processes large "documents" of many cells at once.

**Questions:**

1. In Figure S5, the reconstruction loss for one modality appears unaffected by the presence of the other modality's data. Could the authors clarify how this result aligns with the claim of successful cross-modal information integration?
2. In Figure 4c, the identified "Multimodal Dims." result in lower downstream task accuracy than "Random Dims." Could the authors please explain this counter-intuitive finding?

The reviewer wrote the review. LLM was only used to improve both grammar and clarity.

---

> ### Author Response · Authors · 2025-12-03
> **Response to reviewer DpdC**
>
> We thank you for your thoughtful feedback and for recognizing PETRI as a valuable framework that effectively integrates unpaired multimodal data. We appreciate that you found our interpretability analyses and the release of the matched dataset to be strong contributions. Below, we address your concerns regarding the MST architecture, baseline comparisons, and computational efficiency.
>
> **Role of the Multimodal Set Transformer (MST) and Embedding Quality**
>
> You raised a valid point that embeddings extracted before the MST perform better than those extracted after it, questioning the MST's utility. This phenomenon aligns with findings in self-supervised learning (SSL) literature [1], where "projection heads" or later layers often specialize in the pre-text task (reconstruction) at the expense of general semantic utility.
> In PETRI, the MST acts as the "alignment engine." Its cross-attention mechanism forces the upstream encoders (ViT and Perceiver) to produce tokens that are compatible and aligned before they even enter the MST. We have clarified in the manuscript that the MST is the mechanism that drives alignment in the encoders, even if the encoder output itself is the preferred representation for downstream tasks.
>
> **Quantitative Validation of Biological Concepts**
>
> You noted that our interpretability results (e.g., BODIPY reconstruction) rely on specific examples. The challenge here is that the intersection of biological signals between microscopy and transcriptomics is naturally sparse. Our SAE analysis found ~200 shared dimensions out of 15,000, which reflects this reality.
>
> However, to address your request for broader metrics, we have added a quantitative analysis of "Cross-modal attention variance." We calculated the aggregate cross-modal attention weight across the entire validation set. We found that the attention PETRI pays to the other modality is specific and non-trivial compared to a baseline trained on permuted data (Figure 4d). This confirms that the cross-modal dependencies we visualized are a systematic property of the model, not just anecdotes.
>
> **Fairness of CLIP Baseline**
>
> We agree that standard CLIP is ill-suited for unpaired data. However, variants like CellCLIP have applied contrastive learning to sets of cells in similar regimes, which is why we believe it is an appropriate baseline to include. It demonstrates precisely why contrastive methods fail here: they rely on distinguishing positive pairs from negatives. In screening data with subtle phenotypes and low mutual information, this signal is too weak. The failure of CLIP highlights the necessity of our reconstruction-based approach, which does not require "hard" negatives to learn useful features.
>
> **Computational Cost**
>
> We would like to clarify that PETRI is actually quite efficient. Due to our token resampling strategy, a "document" of 16 cells per modality results in a sequence of only 256 latent tokens (16 cells x 2 modalities x 8 tokens/cell). This sequence length is negligible for modern transformers. Training the full model on the HepG2 dataset took approximately 2 days on a single node of 8 H100 GPUs, which is comparable to standard unimodal training.
>
> **Response to Questions**
>
> - **Figure S5 (Reconstruction Loss)**: You are correct that the average loss doesn't change much. This is because the shared information is sparse (as noted above). The targeted ablation in Figure 4a was necessary precisely because the global average drowns out the specific, local improvements (like predicting lipid droplets from gene counts).
> - **Figure 4c (Multimodal vs. Random Dims)**: The lower accuracy of "Multimodal Dimensions" in predicting the experimental well (batch) is a positive result. It indicates that these dimensions are encoding true biological signal rather than technical batch effects (which are modality-specific). Random dimensions, by contrast, pick up more strongly on the dominant noise/batch signal. We have clarified this section so this interpretation is more explicit.
>
> [1] Guillotine regularization: Why removing layers is needed to improve generalization in self-supervised learning. Bordes, F., Balestriero, R., Garrido, Q., Bardes, A., & Vincent, P. arXiv preprint arXiv:2206.13378, 2022.

---

### Official Review · Reviewer_hVW4 · 2025-10-31

**Soundness:** 3
**Presentation:** 3
**Contribution:** 2
**Rating:** 2
**Confidence:** 4

**Summary:**

Learning and aligning multiple biological modalities in a shared latent space is challenging due to the absence of cell-level pairing and overlap in signals. The paper aims to address this problem by proposing PETRI, an early-fusion transformer model that learns unified cell embeddings of mult-omic modalities in a shared latent space. Cellular images and gene expressions are unpaired and processed as cell documents and learned using a set transformer architecture. Per-cell token resampling is employed and learning is carried out using the MAE reconstruction objective. PETRI, when compared to unimodal and multimodal alternatives, presents imporved GC and StringDB on HepG2 and Perturb-Multi datasets. An interpretable analysis demonstrates that PETRI utilizes cross-modal information while capturing morphological activating features at cell-level resolution.

**Strengths:**

* The paper is well written and organized.
* The paper presents an intuitive perspective on multi-omic learning.

**Weaknesses:**

* **Motivation and Contribution:** The paper motivates multimodal learning of biological modalities using overlap of learning signals and absence of cell-level pairings. However, contributions do not study these aspects and move tangentially towards learning and outperforming prior architectural variants. It is unclear as to how PETRI mitigates conflicting biological signals using a joint representation space. Similar to prior methods, PETRI also learns cell-level embeddings using a masked variational inference objective. This would indicate similar learning signals propogating across modalities. How does the joint space learn and capture cross-modal features? Does PETRI mitigate conflicting gradients or it fits towards a particular modality? How are cell documents arranged in the embedding space given that these are mapped from two different modules (ViT and Perceiver)? In its current form, it remains unclear on how PETRI is addressing the central problem.

* **Empirical Evaluation:** Authors execute experiments on HepG2 and Perturb-Multi datasets which make use of cellular images and transcriptomics profiles. However, the work proposes PETRI from a general multi-omics multimodal learning perspective. Empirical evaluations do not capture this general claim. What happens if we consider additional biological modalities such as genomics and phenomics? Furthermore, experiments do not study the effect of each modality and the contributions presented in the work. How does PETRI behave when one of the modalities is dropped, i.e- a unimodal PETRI? What happens if token resampling is ablated or ViT and Perceiver blocks are replaced by alternative design decisions? In its current form, the paper trivially applies a transformer alternative to the bimodal problem circumventing its technical details.

* **Baselines:** The paper compares PETRI to unimodal as well as multimodal baselines. However, the comparison is limited in scope. Firstly, among the architectural baselines, majoriry of the models are unimodal (imaging or transcriptomics) which does not put the paper's multimodal contributions in persepctive. Secondly, multimodal baselines are limited to architecture and contrastive learning variants which do not highlight the significance of PETRI. What happens if we compare with multimodal VAEs? How does PETRI compare to a multimodal perceiver? How does PETRI compare to CLIP variants and multimodal diffusion? In its current form, baselines and their significance remain limited.

* **Interpretability:** The paper conducts an interpretability-based analysis on how PETRI captures cross-modal features. However, I am having trouble understanding and drawing conclusions from these results. It is unclear as to how the recnstruction loss indicates cross-modal information. Ideally, the reconstruction objective only indicates whether the decoder can disentangle modality-specific information from the latents in order to obtain the original mapping. How does this indicate use of cross-modal features? Similarly for SAE experiments, it remains unclear as to how differences in the number of activated dimensions indicate intepretable cross-modal features. Essentially, a wider difference indicates that more latent dimensions were activated leading to diversity in the feature set. However, this diversity does not necessarilty correlate with the use of cross-modal representations.

**Questions:**

Refer to weaknesses

---

> ### Author Response · Authors · 2025-12-03
> **Response to reviewer hVW4 - Motivation**
>
> We thank you for finding our paper well-written and organized, and for recognizing the intuitive perspective we offer on multi-omic learning. We appreciate your probing questions regarding the mechanics of the joint embedding space and our baseline choices. Below, we address your concerns regarding the motivation, empirical evaluation, and interpretability of our framework.
>
> **Motivation and Learning in the Joint Representation Space**
>
> You raised an important question regarding how PETRI handles conflicting biological signals and how the joint space is learned. Our approach relies on the premise that while modalities like microscopy and transcriptomics are distinct, they share a "ground truth" biological state induced by the perturbation.
>
> PETRI handles potential conflicts through the properties of masked joint reconstruction. Unlike contrastive learning, which forces a direct mapping or alignment, our objective allows the model to treat the other modality as context. The joint space is learned via cross-attention within the Multimodal Set Transformer (MST). If there is shared signal across modalities, the image latent tokens have an opportunity to leverage omics information—backpropagated through the MST— to improve image patch reconstruction (and vice versa).  We see exactly this in our experiments of Fig. 4 where omics tokens aid in reconstruction of masked BODIPY image patches.  On the other hand, if the transcriptomic context conflicts with the image features (or provides no mutual information), the attention mechanism allows the model to separate the tokens and rely on self-reconstruction.
>
> This is empirically supported by our permuted data experiment: when the "aligning" signal of the matched perturbation is broken, the model effectively encodes the modalities disjointly rather than forcing a destructive merge. In particular, the BODIPY reconstruction results for “PETRI permuted” are no worse than a model that ignores omics tokens. This is a positive result indicating that the image tokens have not been “contaminated” by mis-aligned omics signal.
>
> Regarding your question on how documents are arranged given the different encoders, put simply, the "document" is just a sequence of embedding tokens. Regardless of whether they originate from the ViT or Perceiver, we project them to the same number of latent tokens (L) and dimensions (D). These are concatenated into a single sequence of $(2 \times S \times L \times D)$, where S is the number of cells per modality in the document and 2 reflects the two modalities; the cell-specific tokens from each modality can be distinctly separated. The MST then harmonizes these via attention, learning to associate perturbation signatures across the modalities.

---

> ### Author Response · Authors · 2025-12-03
> **Response to reviewer hVW4 - Methods and evaluations**
>
> **Empirical Evaluation and Modalities**
>
> We appreciate your suggestion to explore additional modalities. We softened our claims to make it clear that PETRI is designed only for High-Content Screening (images) and Perturb-seq (transcriptomics). The “document” creation idea used by PETRI may generalize to other settings with unpaired data, but that suggestion is left as follow-up work mentioned in the Conclusion.
> Regarding unimodal performance, we did evaluate "unimodal PETRI" variants (labeled "PETRI Omics" and "PETRI Image" in Table 1). While token resampling does involve a trade-off in information loss, it is necessary to make the joint modeling of cell sets computationally feasible. However, the full PETRI model outperforms both the unimodal baselines and these unimodal variants, motivating this trade-off. We believe the use of standard "off-the-shelf" components (ViT and Perceiver) is a strength, demonstrating that our method of context-based integration is robust without requiring custom architectures. Additionally, our proposed approach for early fusion provides a plug-and-play framework for considering alternative modality-specific encoders.
>
> **Baselines and Related Work**
>
> You asked how PETRI compares to multimodal VAEs or multimodal diffusion.
>
> - **Multimodal VAEs**: Standard multimodal VAEs (like MultiVI, for example) generally require cell-level pairing to model a joint distribution. Since our data is unpaired (different cells, same perturbation), they are not directly applicable. Cross-modal autoencoders are designed for unpaired multimodal data. They are trained in two stages: the first learns a latent space for images and the second learns a latent space for transcriptomics that is aligned to the image space. We have added cross-modal autoencoders as another baseline in the paper and show that it fails to match the performance of PETRI models (Table 1).
> - **Diffusion**: Multimodal diffusion is primarily designed for generation, not representation learning, to our knowledge there are no multimodal diffusion models that work with unpaired data aside from domain translation tasks.
>
> **Interpretability and Cross-Modal Features**
>
> We would like to clarify the reconstruction loss ablation, specifically regarding the BODIPY channel. You questioned how reconstruction indicates cross-modal features. In our experiment, we masked lipid droplets (BODIPY) in the images. We found that the reconstruction error for these specific patches changed depending on the transcriptomic context provided. When we provided omics from a relevant control perturbation, the model could more accurately predict the BODIPY intensity; when we provided omics from a different perturbation, the prediction changed. This proves that the decoder uses cross-modal context to predict masked information.
>
> Regarding the Sparse Autoencoder (SAE) analysis, we define "multimodal dimensions" as those activated by at least 10% of cells in both modalities (see Fig. 5b). This is not a measure of diversity in the latent space, but an indication that the latent space has more shared components when input modalities are properly organized by perturbation. The increased number of these dimensions in PETRI compared to baselines clearly shows that the model learns multimodal concepts.

---

### Meta-Review · Area_Chair_KQ55 · 2026-01-07

**Summary:**

PETRI proposes an early-fusion transformer that learns unified cell embeddings from unpaired microscopy images and single-cell transcriptomics by grouping cells into perturbation-context “documents” and training with masked joint reconstruction + cross-modal attention, enabling simple perturbation-level profiles by averaging embeddings and post-hoc biological concept discovery via sparse autoencoders.

PETRI tackles a real and increasingly important problem in high-content screening: learning a unified representation from unpaired cellular images and transcriptomics where only perturbation context is shared. The paper’s main conceptual contribtuion is to form perturbation-level “multimodal documents” and train an early-fusion transformer with masked joint reconstruction. Improvements on Guide Consistency and StringDB, plus interpretability analyses (SAE concepts, cross-modal reconstruction ablations), indicate a significant advancement in solving this problem. The dataset release is a meaningful community contribution.

The reviews are split. One reviewer is strongly positive (8) and values the dataset, architectural design for unpaired context, and thorough benchmarking; another is mildly positive (4) but raises a serious technical concern that the best embeddings come before the fusion MST, questioning the practical benefit of the central fusion module. Two reviewers are negative (both 2, one with very high confidence), primarily on (i) novelty (components are standard; novelty concentrated in “document construction + token resampling”), (ii) baseline completeness/fairness (unpaired setting limits direct applicability of some methods; CLIP fairness questioned), (iii) evaluation breadth and statistical rigor, and (iv) whether interpretability evidence is systematic vs anecdotal.

The rebuttal has addressed these points of criticism convincingly and improved the paper.
The scope has been narrowed (HCS + Perturb-seq), cross-modal autoencoders as a late-fusion baseline have been added, multi-seed results (N=3) have been included, and quantitative cross-modal attention statistics (significant vs permuted controls) have been introduced. Regarding the role of the MST module, the authors have argued that MST drives alignment pressure into the encoders even if encoder outputs are preferable representations.

Conditional recommendation: lean toward weak accept / poster.

**Reviewer Concerns:**

1. Core motivation/central problem not convincingly addressed: how does PETRI handle partial overlap and potentially conflicting biological signals in a joint space? [hVW4, Gmu9] .
Mostly addressed (conceptually + one key empirical argument). Authors explain that masked joint reconstruction + cross-attention lets modalities act as context rather than forcing alignment; when signals conflict, attention can effectively ignore the other modality. They cite a permuted (mismatched) context experiment showing no “contamination” (performance no worse than ignoring omics), supporting the claim that PETRI doesn’t destructively merge misaligned signals.


2. Generalization claims too broad (multi-omics); limited modality coverage and missing modality-drop analyses/ablations [hVW4] . Mostly addressed. Authors soften scope claims: PETRI is designed for HCS images + Perturb-seq. They also point to “unimodal PETRI” variants (“PETRI Omics/Image” in Table 1), but hVW4’s request for deeper ablations (token resampling, encoder swaps) is only partially answered (they frame off-the-shelf encoders as a strength, and say alternative encoders are plug-and-play).


3. Baselines incomplete: need comparisons to multimodal VAEs/cross-modal generative alignment, multimodal perceiver, diffusion, etc. [hVW4, Gmu9, DpdC] .
Mostly addressed (for the most actionable pieces). Authors add cross-modal autoencoders as a late-fusion baseline and report it underperforms and is unstable. They argue paired-required models like MultiVI aren’t applicable under their unpaired setting. They also defend why diffusion is mostly generative and lacks unpaired biological analogs. This strengthens baseline coverage, though some reviewers may still want additional modern multimodal baselines if they exist for unpaired biology.

4. Interpretability claims are unclear: reconstruction loss doesn’t necessarily imply cross-modal feature use; SAE activation arguments unclear [hVW4, DpdC, Gmu9] .
Authors clarify the targeted reconstruction ablation (masking BODIPY lipid droplets) where predictions change with transcriptomic context, arguing this shows cross-modal use. They also define “multimodal dimensions” more precisely and add quantitative cross-modal attention statistics.

5. Counterintuitive result: embeddings before the fusion block (MST) work better than after; MST seems to hurt downstream metrics [DpdC]. Mostly addressed with plausible explanation, but not a full fix. Authors argue MST behaves like an SSL “projection head” specializing for reconstruction; MST is still useful because it drives alignment pressure into the encoders, and encoder outputs are the best downstream representation. This is coherent, but it’s still somewhat unsatisfying if the “fusion module” degrades downstream evaluation and if no ablation shows MST is necessary for achieving encoder alignment.

6. CLIP baseline potentially unfair given unpaired data; adaptation might handicap CLIP [DpdC, Gmu9] .
Mostly addressed. Authors concede standard CLIP is ill-suited for unpaired settings but argue set-based “CellCLIP-like” variants motivate inclusion, and that CLIP failure illustrates why reconstruction is preferable when mutual information is weak. This is a reasonable defense; fairness still depends on how carefully the CLIP adaptation is implemented and described.

7. Statistical rigor missing (no error bars / multi-seed runs); unclear significance; limited scalability/sensitivity studies [Gmu9] .
Mostly addressed. Authors add N=3 seeds for HepG2 and mention sensitivity analyses in Fig S4. (N=3 may still be viewed as minimal.)

8. Metrics may not capture “biological utility” or multimodal alignment; need clearer downstream use cases beyond embedding proxies [Gmu9] .
Partially addressed but general problem of the field. Authors argue Guide Consistency and StringDB are standard proxies and that cross-modal retrieval can be unhelpful when mutual information is sparse; they position StringDB as evidence for functional gene grouping and suggest batch correction/hypothesis generation via SAE.

9. Computational cost is high [DpdC] .
Authors argue token resampling makes sequences short (e.g., 16 cells/modality → 256 tokens) and report training took ~2 days on 8×H100. That clarifies feasibility in a well-resourced setting, but it does not refute “expensive” in absolute terms; it reframes it as comparable to standard large-scale training.

10. Related work citations missing / context and terminology (“unpaired” vs weakly paired; blinded dataset release) [WChQ].
Authors have incorporated the suggested citations, clarify “blinded” (identities anonymized; structure preserved; some biological validation not reproducible without unblinding), and justify “unpaired” terminology while acknowledging “matched context” framing.

**Reviewer Scores:**

* hVW4: likely 2 → 6. The rebuttal substantially clarifies the core mechanism and adds/justifies baselines; dissatisfaction about depth of ablations/novelty could remain, but the response is materially stronger than before.
* DpdC: likely 4 → 6. The quantitative attention analysis + clarifications on S5/4c address key confusions; MST explanation may be “good enough” for a marginal accept.
* WChQ: likely 8 → 8 (already strongly supportive; rebuttal resolves minor concerns).
* Gmu9: likely 2 → 4 (at best). The rebuttal directly addresses several hard critiques (variance, attention quantification, added baseline), but given confidence 5 and novelty/utility skepticism, a jump to 6 seems unlikely. The AC still believes there are sufficient arguments for acceptance.

---

### Decision · Program_Chairs · 2026-01-26

Accept (Poster)